# Soy-Derived Equol Induces Antioxidant Activity in Zebrafish in an Nrf2-Independent Manner

**DOI:** 10.3390/ijms23095243

**Published:** 2022-05-08

**Authors:** Asami Watanabe, Kyoji Muraki, Junya Tamaoki, Makoto Kobayashi

**Affiliations:** 1Department of Molecular and Developmental Biology, Faculty of Medicine, University of Tsukuba, Tsukuba 305-8575, Japan; watanabe.asami.sg@alumni.tsukuba.ac.jp (A.W.); inter716suarez@gmail.com (K.M.); s1930417@s.tsukuba.ac.jp (J.T.); 2Japan Society for the Promotion of Science (JSPS), Tokyo 102-0083, Japan

**Keywords:** antioxidant effect, arsenite, CRISPR-Cas9, dietary phytochemicals, equol, Keap1-Nrf2 pathway, soy-derived isoflavones, zebrafish

## Abstract

Antioxidant effects of soy-derived isoflavones are predicted to be mediated by the Keap1-Nrf2 pathway. Recently, we constructed an assay system to evaluate the antioxidant effects of dietary phytochemicals in zebrafish and revealed a relationship between these effects and the Keap1-Nrf2 pathway. In this study, we used this system to examine the antioxidant effects of seven isoflavones. Among those seven, equol showed strong antioxidant effects when arsenite was used as an oxidative stressor. The antioxidant effect of equol was also shown in Nrf2-mutant zebrafish *nfe2l2a^fh318^*, suggesting that this effect was not mediated by the Keap1-Nrf2 pathway. To elucidate this unidentified mechanism, the gene expression profiles of equol-treated larvae were analyzed using RNA-seq and qRT-PCR, while no noticeable changes were detected in the expression of genes related to antioxidant effects, except weak induction of Nrf2 target genes. Because *nfe2l2a^fh318^* is an amino acid-substitution mutant (Arg485Lue), we considered that the antioxidant effect of equol in this mutant might be due to residual Nrf2 activity. To examine this possibility, we generated an Nrf2-knockout zebrafish *nfe2l2a^it321^* using CRISPR-Cas9 and analyzed the antioxidant effect of equol. As a result, equol showed strong antioxidant effects even in Nrf2-knockout larvae, suggesting that equol indeed upregulates antioxidant activity in zebrafish in an Nrf2-independent manner.

## 1. Introduction

The beneficial effects of soy foods on health, such as anti-obesity, anti-cancer, and anti-diabetic effects, have been reported [1]. Among the ingredients contained in soy foods, isoflavones have attracted attention as an ingredient responsible for these health-promoting effects. One of the benefits of soy isoflavones is their antioxidant activity; however, this important effect has not been fully investigated [2]. Among the possible candidate antioxidant-mediating biological pathways, the Keap1-Nrf2 pathway has recently been considered a major mediator of the antioxidant effects of soy isoflavones [3,4,5], though it is not clear enough because these studies did not include genetical evidence using Nrf2-knockout mice or alternative animals.

Nrf2 is a master transcription factor that induces expression of antioxidant genes in response to oxidative stress [6,7] and is highly conserved among vertebrates [8]. An important aspect related to preventive medicine is that antioxidant activity in our body can be enhanced by consuming foods containing Nrf2-activating compounds [9]. A well-known Nrf2-activating dietary phytochemical is broccoli-sprout-derived sulforaphane, which has been demonstrated, in murine models to be effective against lifestyle-related diseases [10,11,12]. Nrf2-dependent antioxidant effects of sulforaphane have also been shown in zebrafish [13,14,15]. Zebrafish larvae are especially useful for analyzing antioxidant foods because they hatch within 3 days post fertilization (dpf) and soon begin eating, and their body size (2 mm) is small enough to analyze in multi-well plates. Thus, we recently established an assay system to evaluate the antioxidant effects of food-derived phytochemicals using zebrafish larvae [16]. Using this system, we identified Nrf2-dependent antioxidant effects in five spice-derived phytochemicals (curcumin, diallyl trisulfide, quercetin, isoeugenol, 6-(methylsulfinyl)hexyl isothiocyanate) and an Nrf2-independent effect in cinnamaldehyde. In this study, we analyzed isoflavone compounds and identified equol as an effective Nrf2-independent antioxidant using Nrf2-mutant line *nfe2l2a^fh318^* and a newly generated Nrf2-knockout line *nfe2l2a^it321^*.

## 2. Results

### 2.1. Antioxidant Effects of Isoflavone Compounds in Zebrafish Larvae

Among the isoflavone compounds contained in soy foods, previous reports showed that some isoflavones, such as genistein, have antioxidant activities [17]. In order to examine antioxidant effects of these phytochemicals in zebrafish, we analyzed their protective effects against two different oxidative stressors, hydrogen peroxide and arsenite. We used purified isoflavone compounds (genistin, genistein, glycitin, glycitein, daidzin, daidzein, equol), not extracted isoflavones mixture, and sulforaphane and cinnamaldehyde as positive controls for hydrogen peroxide or arsenite, respectively. Treatment of 2.8 mM hydrogen peroxide killed about 85% of 4-dpf zebrafish larvae in 48 h, while this lethality was drastically reduced when 40 μM sulforaphane was pretreated for 12 h (Figure 1A) [16]. Under this condition, we pretreated isoflavone compounds instead of sulforaphane to test their antioxidant effects at the concentration of 1, 5, and 25 μM. Results showed that neither of the isoflavone compounds at any concentrations alleviated the toxicity of hydrogen peroxide. Treatment of 1.9 mM sodium arsenite killed almost all the 4-dpf zebrafish larvae within 48 h, while this lethality was drastically reduced when 50 μM cinnamaldehyde was pretreated for 12 h (Figure 1B) [16]. Under this condition, 25 μM equol, but not other compounds, could significantly reduce the toxicity of 1.9 mM sodium arsenite treatment. These results indicated that equol showed a strong antioxidant effect in zebrafish when arsenite was used as an oxidative stressor.

### 2.2. Mechanism of Antioxidant Effects by Equol

Because we detected antioxidant effects of equol using our zebrafish assay system, we next investigated its mechanism using zebrafish genetics (Figure 2A). First, we analyzed three concentrations of equol (12.5, 25, and 50 μM) using wild-type larvae to find its optimum concentration and determined that 25 μM was the best. Next, the protective effect of 25 μM equol against arsenite toxicity was analyzed in Nrf2-mutant zebrafish (*nfe2l2a^fh318^*) [13]. The concentration of sodium arsenite was set at 1.4 mM because Nrf2 mutants were weaker to arsenite compared with wild-type larvae [16]. As a result, equol was able to alleviate lethality against arsenite even in the Nrf2-mutant larvae, suggesting that the antioxidant effects of equol were Nrf2-independent.

We next investigated the mechanism of this effect. As isoflavones, including equol, are well known to activate estrogen receptors, we hypothesized that estrogen receptors may mediate the antioxidant effects of equol. To test this hypothesis, we examined the expression of zebrafish estrogen-responsive genes, *vitellogenin 1* (*vtg1*), *estrogen receptor 1* (*esr1*), and *forkhead box C1a* (*foxc1a*) [18,19,20], using quantitative reverse-transcription polymerase chain reaction (qRT-PCR) (Figure 2B). The results showed that the expression of neither gene was significantly induced by equol, suggesting that the antioxidant effect of equol was not based on estrogen-like activity. On the other hand, the expression of *glutathione S-transferase pi 1.2* (*gstp1.2*), a major Nrf2 target in zebrafish [21], showed a weak induction by the equol treatment, though it is not statistically significant (Figure 2C). Because equol has been reported to activate Nrf2 [22,23], we considered that other Nrf2 target genes might also be induced by equol. To examine this possibility, we performed RNA sequence (RNA-seq) analysis to comprehensively investigate Nrf2 target genes using wild-type and *nfe2l2a^fh318^* homozygous larvae treated with 25 μM equol. As a result, 356 and 2580 genes were upregulated (>1.5-fold), and 1598 and 2247 genes were downregulated (>1.5-fold) by the addition of equol in wild-type and *nfe2l2a^fh318^* larvae, respectively (Figure 2D and Appendix A). To identify the biological pathway involved in antioxidant effects, gene ontology analysis was carried out using 73 commonly upregulated and 110 commonly downregulated genes in both wild-type and *nfe2l2a^fh318^* larvae, which are also able to be converted to human homologs (Appendix A). No pathways were found that might contribute to antioxidant effects, including the estrogen receptor and the Keap1-Nrf2 pathways (Appendix A). However, it was noted that many of the Nrf2 target genes showed weak induction, less than 1.5-fold, indicating that the Keap1-Nrf2 pathway might somehow be involved in the antioxidant effect of equol (Figure 2E).

### 2.3. Antioxidant Effects of Equol Were Nrf2-Independent

Because *nfe2l2a^fh318^* is a missense mutant (Arg485Leu) [13], we could not deny the possibility that residual Nrf2 activity in mutant larvae could account for the antioxidant effects. Therefore, we generated a frameshift mutant line *nfe2l2a^it321^*, which was expected to be a null mutant of zebrafish Nrf2, using clustered regularly interspaced short palindromic repeats (CRISPR)-associated protein 9 (Cas9) technology (Figure 3A). The *it321* allele has a 13-base pairs (bp)-insertion in the exon 2 of *nfe2l2a*, resulting in a stop codon insertion in the Neh2 domain of Nrf2. Like the Nrf2-mutant line *nfe2l2a^fh318^*, homozygous *nfe2l2a^it321^* mutants were viable and fertile. To verify the disruption of Nrf2 in *nfe2l2a^it321^* mutants, we analyzed the sulforaphane-induced expression of *gstp1.2* (Figure 3B, left panel). The results showed that the induction observed in wild-type larvae was greatly attenuated to levels comparable to controls in *nfe2l2a^it321^* homozygous larvae. It was noted that the induction in *nfe2l2a^fh318^* larvae was slightly larger than that in *nfe2l2a^it321^* larvae, suggesting that the *nfe2l2a^fh318^* line is a knockdown line with residual Nrf2 activity, as expected. Next, we examined the expression of *gstp1.2* in equol-treated larvae and found that no *gstp1.2* induction was observed in *nfe2l2a^it321^* larvae. Similar results were obtained when we analyzed the sulforaphane-induced expression of an another Nrf2 target gene, *peroxiredoxin 1* (*prdx1*) [24] (Figure 3B, right panel). Based on these results, we considered the newly established *nfe2l2a^it321^* line to be a valid Nrf2-knockout line.

Using *nfe2l2a^it321^* larvae, we next analyzed the protective effects of equol pretreatment against arsenite toxicity (Figure 3C). As a positive control, we used a cinnamon-derived cinnamaldehyde, whose pretreatment had previously shown a strong protective activity against arsenite even in *nfe2l2a^fh318^* [16]. Both compounds could significantly alleviate lethality against 1.4 mM sodium arsenite in *nfe2l2a^it321^* homozygous larvae. The result was similar with that of the Nrf2-mutant line *nfe2l2a^fh318^* (see Figure 2), confirming that the antioxidant effects of equol were indeed Nrf2-independent.

## 3. Discussion

In the current study, we analyzed the antioxidant effects of isoflavone compounds using zebrafish larvae. As a result, we found that equol showed significant antioxidant effects in zebrafish larvae. Moreover, using Nrf2-mutant and newly generated Nrf2-knockout lines, we established that this activity was Nrf2-independent. Equol is one of the isoflavone-derived compounds formed from daidzein by bacteria in animal guts [25]. In the current research, we did not detect the antioxidant effects of daidzein and its glycoside daidzin, suggesting that their metabolism by gut bacteria is important for the acquisition of antioxidant activity, and zebrafish larvae may not have bacteria that can convert daidzein/daidzin to equol. Daidzein has been reported to exhibit antioxidant effects in diet-induced obesity mice [26]. It may be possible that its effect was mediated, at least partially, by its metabolites equol.

Because the antioxidant effects of equol were retained in both Nrf2-mutant and Nrf2-knockout lines, it may be mediated by cellular antioxidant pathways other than the Keap1-Nrf2 pathway. Equol is structurally similar to 17β-estradiol and has a higher estrogenic activity in comparison with daidzein or other isoflavones [27]; further, its antioxidant activity seems to be mostly mediated by its interaction with the estrogen receptors [28]. However, we could not detect equol-induced expression of the typical estrogen receptor target genes, suggesting that the antioxidant effects of equol may not be mediated by the activation of estrogen receptors. Future studies are needed to elucidate the mechanism of equol’s antioxidant effects.

We utilized the Nrf2-mutant line *nfe2l2a^fh318^* extensively. It is a single amino-acid-substituted mutant; nevertheless, it has almost no Nrf2 activity and can cancel the activation of zebrafish Nrf2 target genes via Nrf2-activating compounds [29] and the Keap1 mutation [30]. This indicates the importance of Arg485 in the DNA-binding region. However, the presence of residual Nrf2 activity, albeit weak, was demonstrated in this study. The newly generated Nrf2-knockout line *nfe2l2a^it321^* is a null mutant without Nrf2 activity and is easier to genotype than *nfe2l2a^fh318^* because it is a frameshift mutation with a 13 bp insertion. It will be a useful model for future Nrf2 studies.

## 4. Materials and Methods

### 4.1. Zebrafish and Chemicals

In this study, AB (wild-type), Nrf2-mutant (*nfe2l2a^fh318^*) [13], and Nrf2-knockout (*nfe2l2a^it321^*) zebrafish larvae were used. Both mutant and knockout lines were maintained by PCR-based genotyping. The former was maintained as described previously [14]. For the latter, the primer sets 5′-TATTGTGCAGCCCTAGTGTG and 5′-TAGCTGAAGTCGAACACCTC were used. Larvae used in these experiments were obtained from parents of AB, homozygous *nfe2l2a^fh318^* or homozygous *nfe2l2a^it321^* by natural mating. The *nfe2l2a^fh318^* and *nfe2l2a^it321^* lines can be obtained from the Zebrafish International Resource Center (http://zebrafish.org (accessed on 2022 January 5)) and the National BioResource Project Zebrafish (https://shigen.nig.ac.jp/zebra (accessed on 2022 January 5)), respectively.

Genistein, glycitin, glycitein, daidzin, daidzein, equol, and cinnamaldehyde were purchased from FUJIFILM Wako (Osaka, Japan). Genistin and sulforaphane were purchased from NAGARA Science (Gifu, Japan) and LKT Laboratories (St. Paul, MN, USA), respectively. For stock solutions, hydrogen peroxide and sodium arsenite were dissolved in MilliQ water (Merck-Millipore Billerica, MA, USA), sulforaphane in ethanol, and isoflavone compounds in dimethyl sulfoxide. They were diluted to final concentrations with E3+ medium (5 mM NaCl, 0.17 mM KCl, 0.33 mM CaCl_2_, 0.33 mM MgSO_4_ and 0.1 µg/mL methylene blue).

### 4.2. Survival Assays

Survival assays were performed as previously described [16]. Briefly, eight 3.5-dpf larvae were placed in each well of a 24-well plate with 500 µL of phytochemical solution (E3+ medium containing each phytochemical) for 12 h. At 4 dpf, the phytochemical solution was replaced with oxidative stressor solution (E3+ medium containing hydrogen peroxide or sodium arsenite). The survival of larvae was observed for 48 h after starting the oxidative stressor treatment. Each analysis was performed in triplicate, and the experiments were repeated multiple times to confirm reproducibility. Larvae were not fed during the experiment.

All methods were carried out in accordance with the Regulation for Animal Experiments set out by our University and the Fundamental Guideline for Proper Conduct of Animal Experiment and Related Activities in Academic Research Institutions under the jurisdiction of the Ministry of Education, Culture, Sports, Science, and Technology (MEXT).

### 4.3. Generation of Nrf2-Knockout Line

The *nfe2l2a^it321^* line was generated using CRISPR-Cas9 technology as previously described [31]. In brief, 50 pg of Alt-R crRNA (5′-UGGAUCUGAUCGAUAUCCUGGUUUUAGAGCUAUGCU) (IDT, Coralville, IA, USA) was co-injected with 100 pg Alt-R tracr RNA (IDT) and 400 pg Alt-R Cas9 nuclease V3 (IDT) into the yolk of single-cell-stage wild-type embryos using a microinjector BJ-110 (BEX, Tokyo, Japan). The mutation site was sequenced using the BigDye Terminator v3.1 (Thermo Fisher Scientific, Waltham, MA, USA) and specific primers 5′-TATTGTGCAGCCCTAGTGTG and 5′-TCCTCCGTCTGCTTCAGGTG.

### 4.4. Gene Expression Analyses

For gene expression analyses, total RNA was extracted from 4-dpf larvae treated with sulforaphane or equol for 12 h using ISOGEN II (Nippon Gene, Tokyo, Japan). RNA-seq was performed by Tsukuba i-Laboratory LLP (Tsukuba, Japan) as previously described [30]. A gene ontology analysis was also carried out as previously described [30]. qRT-PCR was carried out as previously described [29]. Primer sets for *vtg1*, *foxc1a*, and *esr1* were as follows: 5′-TGAAAGCAGCAGCAGTCGTA and 5′-ACGAGAGCTGGACATAGAGG (*vtg1*); 5′-CAACATCATGACCTCCCTCC and 5′-TGTTATACCTGTCCGCGAGG (*foxc1a*); 5′-CAGGACCAGCCCGATTCC and 5′-TTAGGGTACATGGGTGAGAGTTTG (*esr1*). Primer sets for *gstp1.2* and *prdx1* were described previously [29].

### 4.5. Statistical Analyses

The survival data were calculated using the Kaplan–Meier method and analyzed with the log-rank test. The statistical significance of gene induction was determined by two-tailed *t*-test. The comparison of gene induction levels between different genotypes was performed using a one-way analysis of variance followed by Bonferroni’s multiple comparisons test. All statistical analyses were performed using EZR, which is a graphical user interface for R (The R Foundation for Statistical Computing, Vienna, Austria).

## Figures and Tables

**Figure 1 ijms-23-05243-f001:**
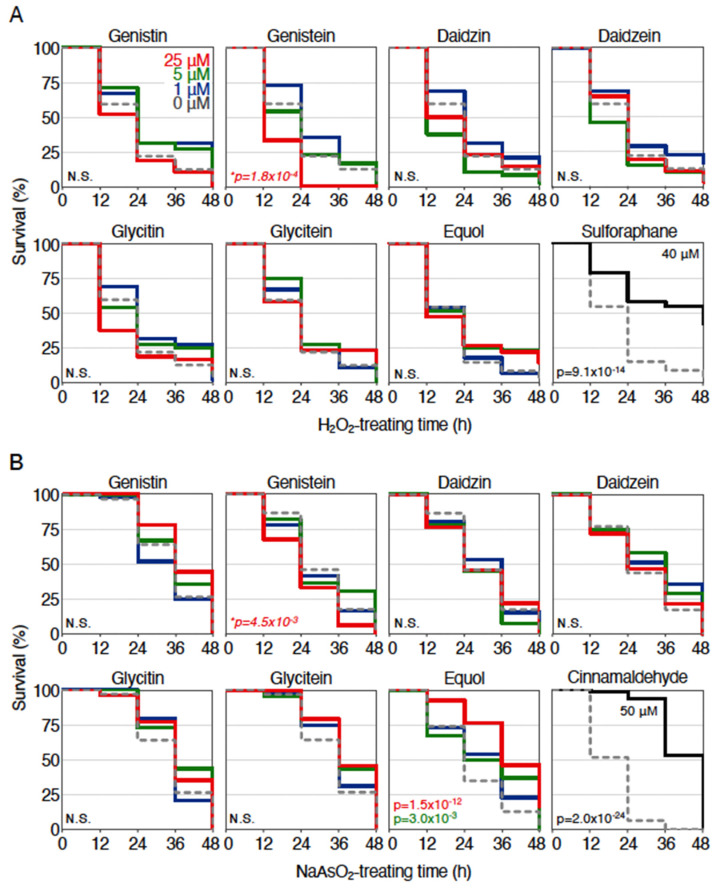
Antioxidant effects of isoflavone compounds in zebrafish larvae. (**A**) Survival assays of hydrogen peroxide (H_2_O_2_)-treated zebrafish larvae using isoflavone compounds. Larvae (3.5 dpf) were pretreated with the indicated isoflavones at concentrations of 0 µM (gray, dotted), 1 µM (dark blue), 5 µM (green), and 25 µM (red). Sulforaphane (40 µM) was used as a control. After pretreatment for 12 h, the solution was changed to 2.8 mM hydrogen peroxide, and survival was measured every 12 h for 48 h. Each analysis was performed in triplicate, and the experiments were repeated multiple times. (**B**) Survival assays of sodium arsenite (NaAsO_2_)-treated zebrafish larvae using isoflavone compounds. They were calculated using the Kaplan–Meier method and analyzed with log-rank test; *p* values of <0.01 were considered to indicate statistical significance. Asterisks denote toxic effects of the indicated phytochemicals. N.S. indicates not significant.

**Figure 2 ijms-23-05243-f002:**
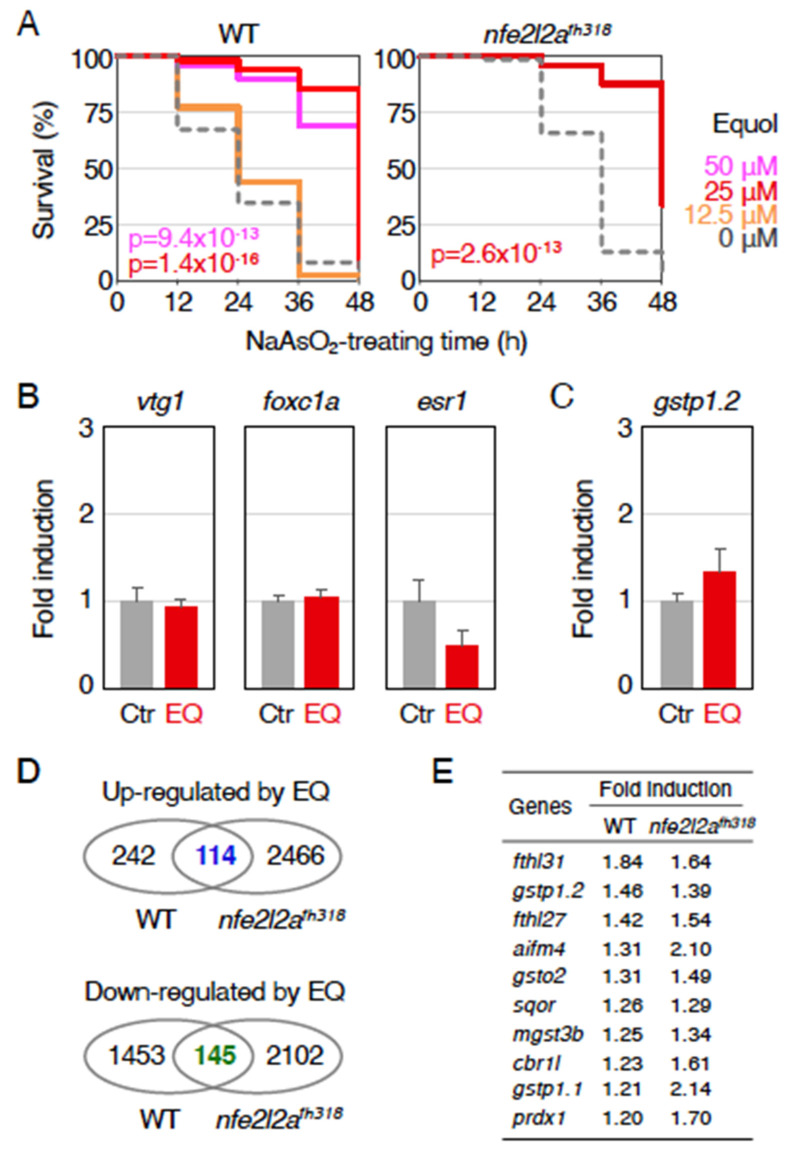
Relationship between equol and the Keap1-Nrf2 pathway. (**A**) Antioxidant effects of equol in Nrf2-mutant larvae. Wild-type larvae (WT) or Nrf2-homozygous mutant larvae (*nfe2l2a^fh318^*) at 3.5 dpf were pretreated with equol at concentrations of 0 µM (gray, dotted), 12.5 µM (orange), 25 µM (red), or 50 µM (pink). After pretreatment for 12 h, the solution was replaced with 1.9 mM (WT) or 1.4 mM (*nfe2l2a^fh318^*) sodium arsenite (NaAsO_2_), and survival was measured every 12 h for 48 h. *P* values of <0.01 were considered to indicate statistical significance. (**B**) Expression of estrogen receptor-target genes in equol-treated larvae. Larvae at 3.5 dpf were treated with or without 25 µM equol (EQ) for 12 h, and the expression of *vtg1*, *foxc1a,* and *esr1* was analyzed using qRT-PCR. The expression of untreated larvae (Ctr) was normalized to 1. Each experiment was conducted at least three times with duplicate samples. (**C**) Equol-induced expression of *gstp1.2*. Larvae at 3.5 dpf were treated with or without 25 µM equol for 12 h, and the expression of *gstp1.2* was analyzed using qRT-PCR. (**D**) Venn diagrams showing upregulated (left) and downregulated genes (right) in 4-dpf larvae treated with 25 µM equol for 12 h identified using RNA-seq analysis. (**E**) List of Nrf2-target genes induced by the equol treatment in RNA-seq analysis. It should be noted that the induction of these genes was not canceled by the *nfe2l2a^fh318^* mutation.

**Figure 3 ijms-23-05243-f003:**
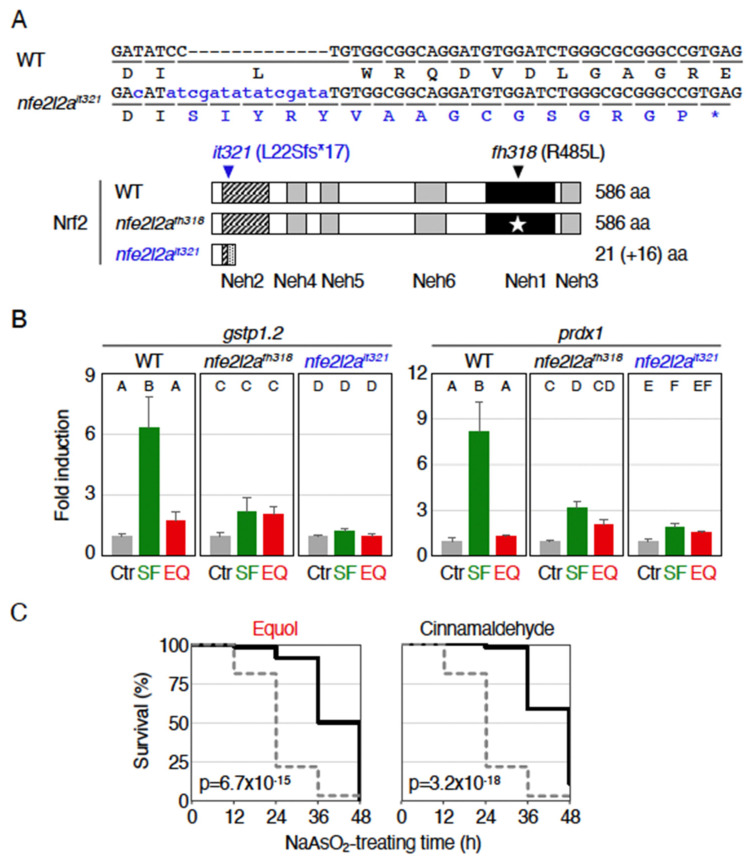
Generation of Nrf2-knockout zebrafish. (**A**) Gene knockout of zebrafish Nrf2 using CRISPR–Cas9 technology. CRISPR target sites were designed in exon 2 of *nfe2l2a* loci (blue arrowhead). In *nfe2l2a^it321^* line, 16-extra-amino-acids were added after the original Ile21 in *nfe2l2a* (diagonal stripes). (**B**) Expression of Nrf2-target genes in sulforaphane- or equol-treated larvae. Larvae at 3.5 dpf were treated without (Ctr) or with 40 µM sulforaphane (SF) or 25 µM equol (EQ) for 12 h in wild-type (WT), Nrf2-mutant (*nfe2l2a^fh318^*), and Nrf2-knockout (*nfe2l2a^it321^*) larvae, and the expression of *gstp1.2* and *prdx1* was analyzed using qRT-PCR. Letters A-F indicate significant differences (*p* < 0.05). (**C**) Antioxidant effects of equol and cinnamaldehyde in Nrf2-knockout larvae. *nfe2l2a^it321^* homozygous larvae at 3.5 dpf were pretreated without (gray, dotted) or with indicated phytochemicals (black, 25 µM equol or 50 µM cinnamaldehyde). After pretreatment for 12 h, the solution was replaced with 1.4 mM sodium arsenite (NaAsO_2_), and survival was measured every 12 h for 48 h. *p* values of <0.01 were considered to indicate statistical significance.

## Data Availability

Not applicable.

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
