# Peer review of "Soy-Derived Equol Induces Antioxidant Activity in Zebrafish in an Nrf2-Independent Manner"

_ijms, 2022, doi:10.3390/ijms23095243_

Round 1

Reviewer 1 Report

Antioxidant effects of soy-derived isoflavones are predicted to be mediated by the Keap1-Nrf2 pathway. In this study, the authors used Nrf2-mutant zebrafish nfe2l2afh318 to examine the antioxidant effects of seven isoflavones. Among those seven, equol showed strong antioxidant effects when arsenite was used as an oxidative stressor. The antioxidant effect of equol was also shown in Nrf2-mutant zebrafish suggesting that this effect was not mediated by the Keap1-Nrf2 pathway. This effect was also confirmed in Nrf2-knockout zebrafish nfe2l2ait321 using CRISPR-Cas9 technology. They concluded that equol upregulated antioxidant activity in zebrafish in an Nrf2-independent manner.

The study is well performed and the system is interesting. The results are clear and convincing. Some minor points should be addressed.

-In lane 80 please specify if Figure 2 is instead Figure 2A. Please amend.

-In lane 97 please insert “Figure 2c”. Please amend.

-The authors should check if the induction of gstp1.2 is statistically significant. Please amend.

-Lane 126-128: “Next, we examined the expression of gstp1.2 in equol-treated larvae, and found that no gstp1.2 induction was observed in nfe2l2ait321 larvae. Similar results were obtained when we analyzed the sulforaphane-induced expression of an another Nrf2 target gene, 128 peroxiredoxin 1 (prdx1)”. Where are these data shown? Please check and amend.

-In Figure 3C survival is shown while it is claimed in the text that antioxidant effect was analysed. Please check and amend.

-The authors show survival claiming antioxidant effect of equol. It would be nice to add an experiment checking the antioxidant effect of equol by analyzing for instance ROS.

-In Materials and Methods please specify the origin of the zebrafish larvae.

-In materials and Methods please describe briefly the survival assay.

Author Response

To Reviewer 1
> The study is well performed and the system is interesting. The results are clear and convincing.
Thank you for your very positive comment.
> -In lane 80 please specify if Figure 2 is instead Figure 2A. Please amend.
We agree with you and replaced "Figure 2" with "Figure 2A".
> -In lane 97 please insert “Figure 2c”. Please amend.
Thank you for pointing our mistake. We inserted "(Figure 2C)".
> -The authors should check if the induction of gstp1.2 is statistically significant. Please amend.
Thank you for your comment. It is not statistically significant. Therefore, we initially considered that equol did not activate the Keap1-Nrf2 pathway. However, when we observed that various Nrf2 target genes were uniformly, but weakly, induced in the RNA-seq analysis, we revised our idea. To avoid readers' misleading, we added a description "though it is not statistically significant" in the results (line 97).
> -Lane 126-128: “Next, we examined the expression of gstp1.2 in equol-treated larvae, and found that no gstp1.2 induction was observed in nfe2l2ait321 larvae. Similar results were obtained when we analyzed the sulforaphane-induced expression of an another Nrf2 target gene, 128 peroxiredoxin 1 (prdx1)”. Where are these data shown? Please check and amend.
We are sorry for the confusing description. The results of prdx1 expression are shown in the right panel of Figure 3B. To avoid readers' misleading, we added descriptions "left panel" (line 123) and "(Figure 3B, right panel)" (line 130) in the results.
> -In Figure 3C survival is shown while it is claimed in the text that antioxidant effect was analysed. Please check and amend.

Thank you for your comment. As you suggested, we replace the description "antioxidant effects of equol pretreatment" with "protective effects of equol pretreatment against arsenite toxicity" in the results (lines 132-133)
> -The authors show survival claiming antioxidant effect of equol. It would be nice to add an experiment checking the antioxidant effect of equol by analyzing for instance ROS.
Thank for your suggestion, we agree with you. We have purchased various "oxidative stress detection probes" and have been trying to detect oxidative stress in zebrafish larvae and embryos, but so far have not been successful. Perhaps these are not sufficient tools to detect oxidative stress in zebrafish in vivo. Also, the small size of zebrafish larvae makes biochemical analysis difficult. We hope to overcome this problem in the
future.
> -In Materials and Methods please specify the origin of the zebrafish larvae.
We are sorry for the confusing description. We replaced the description "Embryos used in these experiments were obtained by natural mating." with "Larvae used in these experiments were obtained from parents of AB, homozygous nfe2l2afh318 or homozygous nfe2l2ait321 by natural mating." (lines 216-217)
> -In materials and Methods please describe briefly the survival assay.
As you suggested, we added descriptions in the Materials and Methods as following: "Briefly, eight 3.5-dpf larvae were placed in each well of a 24-well plate with 500 μl of phytochemical solution (E3+ medium containing each phytochemical) for 12 h. At 4 dpf, the phytochemical solution was replaced with oxidative
stressor solution (E3+ medium containing hydrogen peroxide or sodium arsenite). The survival of larvae was observed for 48 h after starting the oxidative stressor treatment. Each analysis was performed in triplicate, and the experiments were repeated multiple times to confirm reproducibility. Larvae were not fed during the experiment." (lines 228-235)

Reviewer 2 Report

An interesting paper on soy-derived equinol relevance on nrf 2anxioxidant effect in ZF larvae

I think the paper has some interest for the readers

Introduction is good, with current state of the art.

Some parts are still in need of some referencing:

Nrf2-activating dietary phytochemical is broccoli sprout-derived sulforaphane, 45 which has been demonstrated, in murine models, to be effective against lifestyle-related 46 diseases (reference?)

Methodology is very good and advanced.

Still, I would suggest describe the method again (there is enough space on MDPI journals and servers), rather then sending to recently published paper in MDPI also few months ago

Results is very very robust – no suggestions here

Also I find the Discussion and Conclusion section well balanced and ok

Author Response

To Reviewer 2
> An interesting paper on soy-derived equinol relevance on nrf 2anxioxidant effect in ZF larvae. I think the paper has some interest for the readers. Introduction is good, with current state of the art.
Thank you for your very positive comment.
> Nrf2-activating dietary phytochemical is broccoli sprout-derived sulforaphane, which has been demonstrated, in murine models, to be effective against lifestyle-related diseases (reference?)
As you suggested, we added 3 selected references [10-12].
> Still, I would suggest describe the method again (there is enough space on MDPI journals and servers), rather then sending to recently published paper in MDPI also few months ago Reviewer 1 gave us similar suggestions. As you suggested, we added descriptions in the Materials and Methods (lines 216-217, 228-235).